# Enhancing generalizability in classification of peripheral neural recordings with graph neural network

Rui Qi Ji[1]*, Mehdy Dousty[1,2‡], Ryan G. L. Koh[3,4¶], Ervin Sejdić[1,5]

1 The Edward S. Rogers Sr. Department of Electrical & Computer Engineering, Faculty of Applied Science & Engineering, University of Toronto, Toronto, Ontario, Canada, 2 Vector Institute, Toronto, Ontario, Canada, 3 Institute of Biomedical Engineering, University of Toronto, Toronto, Ontario, Canada, 4 KITE Research Institute, Toronto Rehabilitation Institute, University Health Network, Toronto, Ontario, Canada, 5 North York General Hospital, Toronto, Ontario, Canada,

‡ Supported by University of Toronto's Eric and Wendy Schmidt AI in Science Postdoctoral Fellowship.
¶ Supported by University of Toronto's Data Science Institute Postdoctoral Fellowship.
* ruiqi.ji@mail.utoronto.ca

## Abstract

The peripheral nervous system plays a crucial role in facilitating communication between biological systems. However, decoding neural signals from peripheral nerve recordings remains a challenge due to their complex spatiotemporal patterns. In this study, we propose a graph-based learning approach to more effectively capture temporal and spatial information for classifying neural signal patterns. Unlike previous work, our method incorporates the physical geometry of the nerve cuff, addressing the underrepresented relationships between electrodes. We used a publicly available dataset consisting of neural recordings from eight Long-Evans rats, obtained using a 56-channel nerve cuff electrode. We constructed graphs where each node represents the time series recorded from an electrode, and edges correspond to the distances between electrodes along the surface of the nerve cuff (e.g., geodesic distance). We employed a leave-one-out strategy to evaluate the generalizability of the approach. We further evaluated the within-rat performance of the model by training on two folds and testing on the remaining fold of each rat's data. In generalizability evaluation, we achieved a mean F1 score of 65.03%, representing a 17.74% improvement over the previous study, and in within-rat testing, we achieved a mean F1 score of 77.50%, representing a 3.14% increase. These findings highlight the value of incorporating the recording geometry into model design, particularly in this small dataset setting, where explicit spatial priors help compensate for limited training examples and improve decoding performance.

**Data availability statement:** The data used in this study were from a publicly available dataset available at Borealis, U of T Dataverse: https://doi.org/10.5683/SP3/JRZDDR.

**Funding:** The author(s) received no specific funding for this work.

**Competing interests:** The authors have declared that no competing interests exist.

## Introduction

The nervous system is a fundamental component of biological function, serving as the primary communication network within the body [1]. It is responsible for transmitting electrical and chemical signals that regulate movement, sensation, and autonomic processes [2,3]. The peripheral nervous system facilitates bidirectional communication between the brain and the rest of the body [4,5]. As such, peripheral neural signals encode crucial information about sensory inputs and motor commands, making their accurate interpretation essential for various biomedical engineering applications, and the diagnosis of neurologic disorders [6–8].

Nevertheless, obtaining reliable and selective recordings in the peripheral nervous system is challenging [9]. Recent studies have utilized multi-contact electrodes [10–12] to achieve encoding both temporal and spatial information of neural recordings in Long-Evan rats. Convolutional neural networks (CNNs) have demonstrated significant capability in analyzing physiological data [13–16]. Koh et al. proposed a CNN-based approach to classify three afferent activities in peripheral neural recordings from Long-Evans rats: dorsiflexion, plantarflexion, and pricking [12]. These movements are fundamental to locomotion, balance, and pain perception [17]. Thus, classifying these three types of neural activity is critical for understanding motor control mechanisms. Such classification has direct implications in neuroprosthetics, rehabilitation, and sensory feedback systems for amputees or patients with neuromuscular disorders [17,18].

CNNs operate through receptive fields and therefore capture the structural layout of multichannel neural recordings, enabling them to model local spatial dependencies by leveraging weight sharing [19]. Transformers, based on self-attention, have been shown to lack the strong architectural inductive biases of convolutional networks, such as limited receptive fields and translational invariance, that can be beneficial for spatially structured inputs [20]. Although Transformers can model long-range dependencies [21], they require substantially more data to learn useful representations in the absence of such biases, and do not inherently encode the geometric relationships present in nerve cuff recordings. Similarly, other neural network architectures commonly applied in the literature for peripheral neural signal analysis or classic machine learning classifiers do not encode the true inter-electrode distances or circumferential arrangement around the nerve, meaning that important physiological dimensions of electrode placement, such as the true inter-electrode distances and the way neural signals propagate across adjacent contacts, remain unmodeled. In contrast, graph neural networks (GNNs) offer greater flexibility in representing irregular topologies, as they enable a more adaptive encoding of both temporal and spatial relationships in neural data [22]. Unlike CNNs, which require large amounts of data to learn spatial filters on fixed grid structures, GNNs incorporate structural priors through graph connectivity, enabling more effective biosignal analysis and improved generalization in data-limited settings [23–27]. Previous work has also highlighted the importance of how combining both temporal and spatial information yields the highest performance [28], thus, the main motivation for using GNNs in our context is their ability to simultaneously encode the data's temporal and structural relationships. For instance,

electrodes located on opposite sides of a nerve cuff may record similar physiological activity, yet this relationship cannot be effectively captured using conventional CNN or Transformer architectures. By modeling these electrode connections as edges in a graph, GNNs naturally integrate this spatial context alongside temporal dynamics.

In this paper, we propose a graph-based learning approach for classifying peripheral neural signals. Our key contributions are

1) **Geometry-aware graph construction:** We model nerve cuff electrodes as graph nodes and define edges based on geodesic distances to encode both the physical arrangement of electrodes and functional similarities between neural signals. We examine how varying graph connectivity affects model performance and conduct ablation studies to verify that the proposed graph-based approach is the primary contributor to the performance gains. In the ablation studies, we also compare the feature extraction from neural recordings using a LSTM module versus a 1D CNN module.

2) **Improved generalizability in small-data regimes:** Our approach achieves higher classification accuracy and better generalization, outperforming the CNN baseline from previous studies and highlighting the importance of encoding information with graphs in smaller datasets.

## Materials and methods

### Data description & preprocessing

Neural recordings were previously collected from nine Long-Evan rats from the sciatic nerve using a 56-multi-contact nerve cuff electrode were used [11]. In this study, we excluded one rat due to an issue with the degradation of the plantarflexion signal, resulting in a dataset comprising of eight rats. Neural recordings were collected from a 56 channel nerve cuff electrode comprised of 7 rings of 8 contact, evenly distributed over the length of the electrode. The recordings were acquired at a sampling frequency of 30 kHz with a neural data acquisition board (RHD2000, Intan Technologies, USA) [11]. Three afferent activities, dorsiflexion, plantarflexion, and pricking, were manually performed to evoke neural activity in the recording. Naturally evoked compound action potentials (nCAPs), produced by proprioceptive or mechano-sensory afferent activity in response to physiological limb movements or mechanical stimulation, were detected, and used to construct spatiotemporal signatures for each activity. Each spatiotemporal signature is a matrix in which rows represent neural activity of individual channels over time, and columns capture signals at specific time points across channels, resulting in a matrix of size 56 by 100 (number of contacts by time samples) [11,12,29]. These spatiotemporal signatures were then constructed into graph structures, which were fed into our proposed model for classification of the three activities. Representing the data as graphs allows us to explicitly encode spatial relationships between electrodes, which may capture physiologically meaningful similarities that conventional matrix-based approaches overlook. The number of samples for each rat is presented in Table 1.

### Graphs & adjacency matrices

A graph can be represented mathematically as an order pair $G=(V,E)$, where $V$ represents the nodes, and $E$ represents the edges, connecting pairs of nodes together [30]. The graph can be mathematically expressed through an adjacency matrix A, where $A_{ij}$ indicates the presence of an edge between the nodes. If only the existence of edges is considered, the graph is unweighted, where $A_{ij}=1$ if the edge exists between $i$ and $j$, and $A_{ij}=0$ otherwise. In weighted graphs, non-zero values of $A_{ij}$ also represent the strength or significance of the connection [31]. GNNs are deep learning architectures designed to operate directly on graph-structured data. Unlike conventional models that assume regularly structured 1D, 2D, or 3D inputs, GNNs leverage both node features and the connectivity defined by edges to learn representations that capture the underlying topology [22]. Through iterative message passing, each node aggregates information from its neighbors, enabling the model to integrate both local and global structural patterns in the data.

Table 1. Number of samples from each Long-Evan rat, and number of samples for each class.

| Rat # | Number of Samples | Dorsiflexion | Plantarflexion | Pricking |
|-------|-------------------|--------------|----------------|----------|
| Rat 2 | 34,238 | 4,044 | 13,272 | 16,922 |
| Rat 4 | 31,346 | 8,602 | 12,248 | 10,496 |
| Rat 5 | 37,610 | 10,088 | 9,948 | 17,574 |
| Rat 6 | 29,878 | 11,744 | 3,704 | 14,430 |
| Rat 7 | 22,506 | 4,438 | 9,640 | 8,428 |
| Rat 8 | 23,532 | 5,208 | 11,872 | 6,452 |
| Rat 9 | 21,396 | 9,358 | 3,306 | 8,732 |
| Rat 10 | 11,174 | 2,600 | 3,004 | 5570 |

In this work, we represent the electrode contact points (i.e., channels) as nodes and define the edges based on the geodesic distance relationships between them. The nerve cuff consists of multiple electrodes arranged circumferentially around the nerve, so electrodes positioned on opposite sides may capture similar neural activity due to their proximity to the same underlying fibers. The geodesic distance measures the shortest path along a curved surface rather than the straight-line Euclidean distance [32]. This distinction is particularly important for nerve cuff electrodes, which are positioned on a cylindrical surface around the nerve. While Euclidean distance may treat two electrodes on opposite sides of the cuff as far apart, geodesic distance captures their true proximity along the nerve's surface, providing a more physiologically meaningful representation of spatial relationships.

Given the structure of the nerve cuff electrode, this distance is computed using Equation 1 below, in which $x$ and $y$ represent the row and column indices (e.g., electrode coordinates) of the electrode channels in the nerve cuff array, respectively. The term $|x_i - x_j|$ captures the vertical distance between electrodes, while $\min(|y_i - y_j|, 8 - |y_i - y_j|)$ accounts for the wrap-around nature of the circular arrangement in the horizontal direction, as shown in Fig 1.

$$d_{\text{geodesic}}(i, j) = |x_i - x_j| + \min(|y_i - y_j|, 8 - |y_i - y_j|) \tag{1}$$

By utilizing geodesic distance, electrodes that are closer to each other on the cuff are more likely to be directly connected in the graph, representing stronger spatial relationships. This connectivity structure enables the model to learn signal propagation patterns along the electrodes, rather than being limited to local information as in other methods, thereby better capturing spatial correlations and neural dynamics within the recordings. With this approach, we constructed a weighted adjacency matrix of size 56 by 56, where the weights are computed based on the distance between electrode positions, measured by their geodesic distance, as shown in Equation 2 [33].

$$weight(i, j) = \exp\left(-\frac{d_{\text{geodesic}}(i, j)^2}{2\sigma^2}\right) \tag{2}$$

To refine the graph topology, we use the distance-scaling parameter $\sigma$ and the number of nearest neighbours, $k$, as tunable hyperparameters. Specifically, for each node, edges are first formed only with its $k$ nearest neighboring electrodes based on spatial distance, and the corresponding edge weights are assigned using a distance-based weighting function parametrized by $\sigma$. With lower values of $\sigma$, only electrodes that are immediately adjacent on the nerve cuff will have substantial edge weights, leading to sharper decay in edge weights and emphasizing local interactions. Larger $\sigma$ values result in high weights even for electrodes located further apart along the cuff, allowing the model to integrate global information from distant regions of the nerve. By restricting connectivity to the $k$ nearest neighbors, the graph remains sparse and focuses message passing on the most relevant spatial relationships, while $\sigma$ controls the strength of these connections.

 

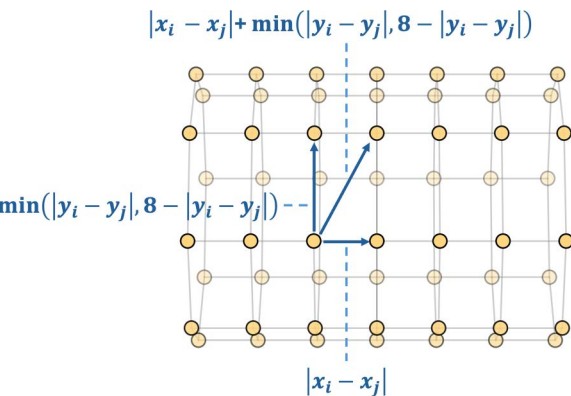

**Fig 1. Geodesic distance configuration in graph construction, where *x* and *y* represent the row and column indices (e.g., electrode coordinates).**

This combined-tuning mechanism enables flexible tuning of the graph structure, balancing the trade-off between local specificity and global connectivity, ultimately improving the model's ability to capture informative patterns from the neural data.

## Model architecture

The model architecture is implemented with a hybrid approach, incorporating both sequential and graph-based learning paradigms. The full model architecture is shown in Fig 2. Following the graph formulation introduced above, each neural recording is represented as a graph $G = (V,E)$, where each node $v_i \in V$ corresponds to an electrode contact and each edge $e_{ij} \in E$ encodes the spatial relationship between electrodes. The graph structure is represented by a weighted adjacency matrix $A$, where each non-zero entry $A_{ij}$ denotes the strength of the connection between nodes $i$ and $j$, as defined by the geodesic distance-based graph construction procedure.

We first extract temporal representation from the neural signals using a Long Short-Term Memory (LSTM) layer with 256 units. The resulting hidden representations serve as node feature vectors $h_i$ for each $v_i \in V$. These node features, together with the weighted adjacency matrix $A$, are then processed by an edge convolution layer followed by a general graph convolution layer.

The edge convolution layer follows a message-passing formulation in which node features are updated by aggregating information from neighboring nodes $j \in \mathcal{N}(i)$, explicitly incorporating edge information through the corresponding adjacency weights $A_{ij}$. In this work, edge features are defined directly by these weighted adjacency values, which encode spatial proximity between electrode contacts. This allows adaptive feature learning that captures spatial dependencies expressed in adjacency matrices [34]. An edge convolution update can be represented by Equation 3, where $h_i^{(l)}$ denotes the feature vector of node $i$ at layer $l$, and $h_j^l$ denotes the feature vector of a neighboring node $j \in \mathcal{N}(i)$. The term $A_{ij}$ corresponds to the weighted adjacency value between nodes $i$ and $j$, which represents edge features and encodes the spatial relationship between electrode contacts based on the distance-based graph construction. The function $\phi(\cdot)$ is a learnable mapping implemented as a multilayer perceptron that combines node features and edge information to generate messages from neighboring nodes.

General graph convolutions then aggregate node features based on neighborhood information, enabling the model to learn feature dependencies across the graph structure, as shown in Equation 4 [35]. Here, $H^{(l)}$ represents the matrix of node features at layer $l$, $\tilde{A} = A + I$ is the adjacency matrix with added self-loops, and $\tilde{D}$ is the corresponding degree matrix. The matrix $W^{(l)}$ contains learnable weights, and $\sigma(\cdot)$ denotes a nonlinear activation function. This formulation enables the

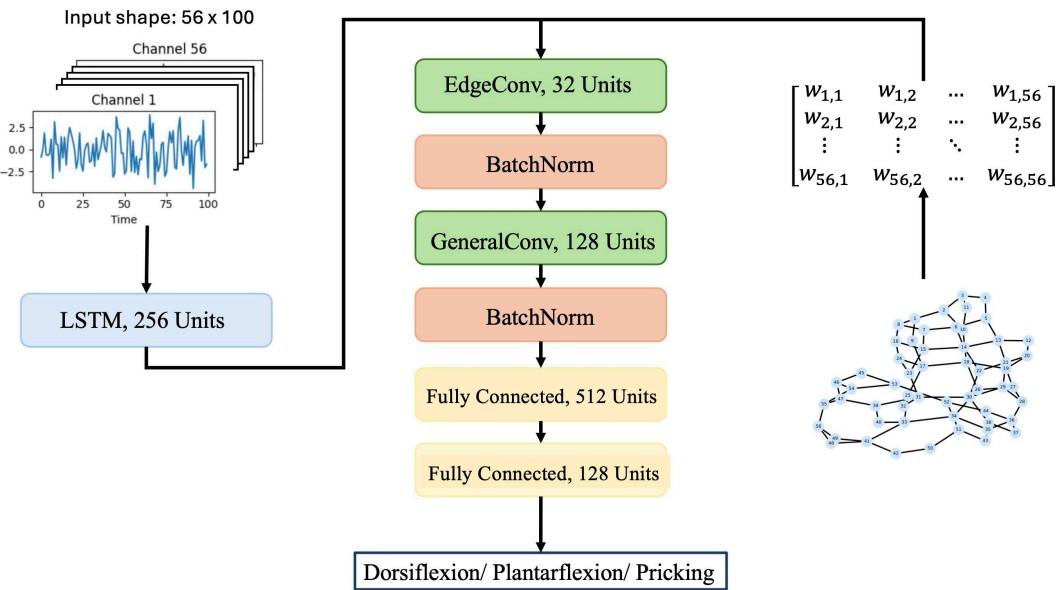

**Fig 2. Proposed model architecture.**

model to propagate and integrate information across the graph while preserving its underlying structure. Together, these convolutional layers ensure that both local and global structural properties of the graph input are effectively processed.

$$\mathbf{h}_i^{(l+1)} = \sum_{j \in \mathcal{N}(i)} \phi\left(\mathbf{h}_i^{(l)}, \mathbf{h}_j^{(l)}, A_{ij}\right),$$

$$\tag{3}$$

$$\mathbf{H}^{(l+1)} = \sigma\left(\tilde{\mathbf{D}}^{-\frac{1}{2}}\tilde{\mathbf{A}}\tilde{\mathbf{D}}^{-\frac{1}{2}}\mathbf{H}^{(l)}\mathbf{W}^{(l)}\right),$$

$$\tag{4}$$

In this model, an edge convolution layer with 32 units and a general graph covolution with 128 units were used, with an L2 regularizer of magnitude $5 \times 10^{-3}$ to improve generalizability. The final feature representation undergoes global average pooling before being passed through fully connected layers with rectified linear unit (ReLU) activations. Finally, we use a softmax layer to output the classification probabilities. During training, a batch size of 1024 and a learning rate of 0.001 were used. A 20% dropout rate is applied to the dense layers to enhance regularization and reduce overfitting.

Prior to training, we applied data augmentation in the form of low-amplitude Gaussian noise, which serves as a physiologically meaningful perturbation for neural recordings. Gaussian noise injection acts as a regularizer by simulating naturally occurring variability in peripheral nerve signals—such as background neural activity, electrode impedance fluctuations, and minor recording noise—while preserving the underlying spatiotemporal structure of the compound action potentials. This approach helps improve model robustness and generalizability without altering the temporal dynamics or spatial relationships that are essential for accurate decoding.

### Training & evaluation

**Hyperparameter tuning.** We tuned the model hyperparameters using data from two randomly selected rats (e.g., Rats 2 and 10), which served as a validation set for hyperparameter optimization. The training hyperparameters,

including the learning rate (1e-4 –1e-2), batch size (128–1024), and L2 regularization coefficient (1e-5 – 5e-3), were systematically tuned based on validation performance. In addition, architectural parameters such as the number of hidden units (128–512), dropout rates (0.1–0.8), and overall model layer configurations were also selected through the same validation-based tuning procedure to balance model capacity and generalization. Once the optimal hyperparameters were determined, these rats were reincorporated into the training and evaluation process to maximize data utilization. In addition, we systematically varied key graph construction parameters, such as the number of neighbors and the distance decay parameter ($\sigma$) in the adjacency matrix, to analyze their impact on model performance. Model development and training were carried out in Python using the TensorFlow deep learning framework.

**Evaluation.** Model performance was evaluated using both cross-subject and within-subject strategies. For generalizability across rats, we employed an eight-fold leave-one-out cross-validation approach, where data from seven rats were used for training and the remaining rat was held out for testing. This design allowed us to rigorously assess across-subject generalization, addressing a limitation of prior CNN-based studies that focused only on within-subject performance [12]. Performance was quantified using test accuracy and the macro-averaged F1 score, which is more reliable for imbalanced datasets. To ensure comparability with previous work, we also conducted within-subject evaluations by performing a cross-validation in which two folds were used for training and the remaining fold was used for testing. This process was repeated such that each fold served as the test set once.

## Ablation studies

To evaluate the contribution of specific architectural and structural design choices in our proposed framework, we also conducted three sets of ablation studies. These studies were aimed at disentangling the effects of (1) temporal modeling via LSTM and (2) graph construction based on geodesic distances. To ensure that the changes in performance were not solely attributable to the use of LSTM, we replaced the it with a 1D CNN module. This design choice also aligns more closely with the architecture used by the previous study [11,12], thereby enabling a more direct comparison with prior work. In addition to the proposed geodesic graph, we constructed graphs based on Euclidean distances between electrode contacts to evaluate whether preserving the true surface geometry of the nerve cuff provides advantages over simpler spatial proximity measures. Euclidean distances were computed directly from the two-dimensional spatial coordinates of the electrode contacts, without accounting for the curved surface geometry of the nerve cuff. Furthermore, we constructed a random graph baseline in which edges were assigned randomly between nodes. This setup preserved the graph's sparsity while removing the physiological prior embedded in the geodesic topology, allowing us to observe the performance of the model under different spatial constraints.

## Results

### Generalizability performance

Table 2 presents the classification accuracies (%) and macro-averaged F1 score for each individual rat when used as a test set, as well as the mean and standard deviation across all rats. The first row shows the performance of the baseline CNN model reimplemented from the previous study [12] and the subsequent rows show the graph-based model proposed in this work. For the graph-based model, we compared how varying the number of neighbouring nodes changes the performance at a fixed $\sigma$ of 2. This allows us to better examine the effects of neighbouring nodes in classification performances. For the graph-based models, we have grayed out the performance scores for Rats 2 and 10, as these subjects were used as a validation set during hyperparameter tuning. Accordingly, their results were excluded from the calculation of the mean ± standard deviation. The scores are nonetheless reported to provide additional context regarding their individual performance. Overall, the graph-based approaches resulted in an improved mean classification accuracy compared to the CNN model, which achieved 54.00 ± 5.21% (* beside the mean ± standard deviation indicates significant difference,

**Table 2. Generalization performance (accuracy and macro-averaged F1 score) of baseline model and graph-based models with varying connectivity (Rats 2 and 10 are excluded from final mean ± standard deviation evaluation as they were used in the validation set for hyperparameter tuning).**

| Metrics | Model | Rat 2 | Rat 4 | Rat 5 | Rat 6 | Rat 7 | Rat 8 | Rat 9 | Rat 10 | Mean ± Standard Deviation |
|---|---|---|---|---|---|---|---|---|---|---|
| Accuracies | CNN | 48.00 | 53.57 | 51.57 | 47.70 | 58.41 | 61.82 | 50.93 | 43.96 | 54.00 ± 5.21 |
| | 2 neighbors | 54.63 | 57.32 | 54.07 | 77.88 | 52.53 | 60.81 | 43.22 | 51.38 | 57.64 ± 11.55 |
| | 3 neighbors | 53.49 | 58.71 | 67.49 | 69.78 | 62.51 | 58.92 | 54.14 | 50.45 | 61.93 ± 5.88 |
| | 4 neighbors | 49.43 | 60.91 | 68.25 | 73.14 | 60.48 | 63.92 | 58.95 | 51.02 | **64.28 ± 5.45\*** |
| | 5 neighbors | 51.71 | 62.89 | 64.98 | 72.90 | 68.65 | 72.71 | 67.78 | 52.17 | **68.32 ± 4.03\*** |
| | 6 neighbors | 52.30 | 60.08 | 62.88 | 68.43 | 65.05 | 56.73 | 68.49 | 53.92 | 63.61 ± 4.68 |
| | 7 neighbors | 50.58 | 58.99 | 66.42 | 64.33 | 56.30 | 69.27 | 37.59 | 49.77 | 58.82 ± 11.44 |
| | 8 neighbors | 50.58 | 40.09 | 53.66 | 59.55 | 48.28 | 69.27 | 43.74 | 49.84 | 52.43 ± 10.78 |
| | 9 neighbors | 38.75 | 55.33 | 55.64 | 66.06 | 30.90 | 56.57 | 41.28 | 49.87 | 50.96 ± 12.62 |
| F1 Score | CNN | 35.99 | 44.67 | 47.22 | 46.55 | 47.09 | 46.44 | 51.76 | 33.34 | 47.29 ± 2.37 |
| | 2 neighbors | 40.38 | 54.45 | 51.77 | 65.43 | 40.27 | 54.50 | 29.29 | 29.69 | 49.29 ± 12.66 |
| | 3 neighbors | 36.08 | 58.62 | 61.40 | 59.82 | 60.80 | 55.52 | 50.77 | 26.91 | **57.82 ± 4.03\*** |
| | 4 neighbors | 22.05 | 60.13 | 56.50 | 66.46 | 58.88 | 61.24 | 55.10 | 27.56 | **59.72 ± 4.01\*** |
| | 5 neighbors | 38.18 | 60.27 | 62.54 | 63.98 | 65.56 | 72.64 | 65.19 | 33.63 | **65.03 ± 4.20\*** |
| | 6 neighbors | 46.57 | 60.19 | 54.80 | 60.80 | 59.86 | 59.43 | 62.86 | 46.00 | **59.66 ± 2.67\*** |
| | 7 neighbors | 40.00 | 49.30 | 61.33 | 57.10 | 48.99 | 60.83 | 27.13 | 23.62 | 50.78 ± 12.78 |
| | 8 neighbors | 41.16 | 35.74 | 53.08 | 53.86 | 36.10 | 70.46 | 27.67 | 24.57 | 46.15 ± 15.81 |
| | 9 neighbors | 39.05 | 55.53 | 55.07 | 49.66 | 29.95 | 56.86 | 23.06 | 24.90 | 45.02 ± 14.71 |

$p < 0.05$ from the baseline model performance, computed using the t-test). The graph-based approach with connectivity defined by four or five neighbors showed a statistically significant improvement from the CNN model in terms of F1-score, and among the graph learning models evaluated, the graphs constructed with five neighbors was the highest-performing model, outperforming the baseline CNN by 14.32% in mean classification accuracy and 17.74% in F1 score. Fig 3 illustrates graph connectivity with five neighbors, which is the optimal connectivity found.

To systematically analyze the effect of graph construction hyperparameters, we present a comprehensive heatmap of averaged F1-scores across all rats while varying the number of neighbors and $\sigma$ (Fig 4). This visualization highlights how model performance changes with different parameter combinations and demonstrates that the chosen hyperparameters ($\sigma$ of 2 and number of neighbors of 5) achieve the highest classification accuracy. We then performed ablation studies with these chosen hyperparameters to evaluate model generalizability, with results summarized in Table 3. Fig 5a and 5b further illustrate the graph connectivity constructed using Euclidean distance and random assignments, respectively. Compared with the geodesic-distance-based connectivity shown in Fig 3, these alternative graph structures exhibit remarkably different topologies.

Substituting the LSTM branch with a CNN did not provide significant performance gains, and our model still outperformed prior CNN-based work. Interestingly, when using a random graph in place of the correct adjacency matrix, the accuracy remained close to that of the CNN baseline. This observation likely reflects the limited generalization capability of the CNN baseline in the across-subject setting. CNNs rely on fixed, grid-based receptive fields and tend to learn subject-specific spatial patterns that do not transfer well across rats. As a result, both the CNN and the random-graph model lack an explicit inductive bias that enforces physiologically meaningful spatial relationships between electrodes, leading to similar generalization performance. In contrast, incorporating anatomically informed graph connectivity enables the GNN to leverage consistent spatial organization across subjects, resulting in substantially improved performance.

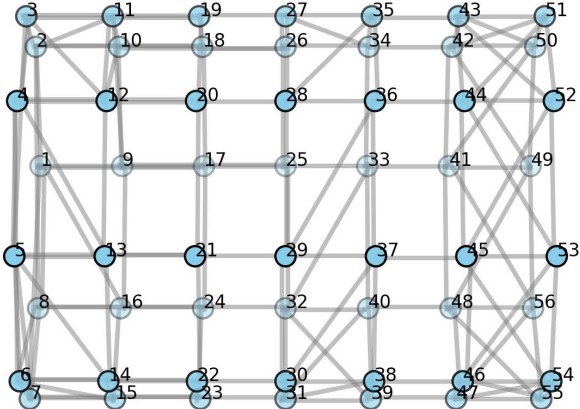

**Fig 3. Graph connectivity with five neighbors, defined by geodesic distance.**

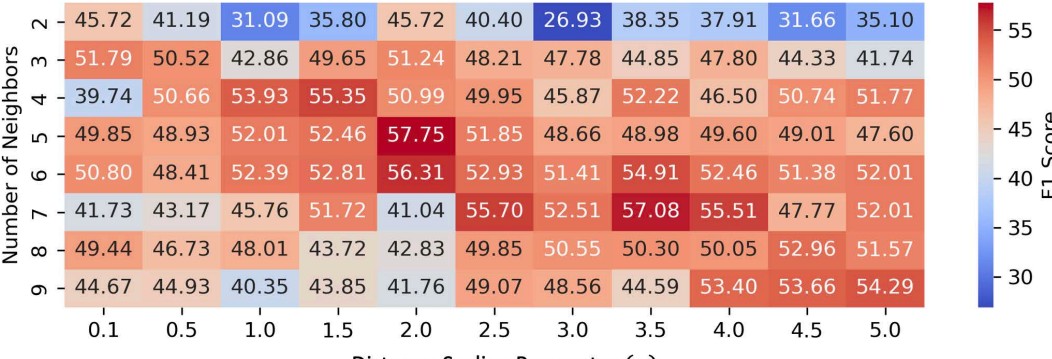

**Fig 4. Heatmap of F1-scores across different distance decay parameters and number of neighbors.**

**Table 3. Generalization performance (accuracy and macro-averaged F1 score) of ablation studies (Rats 2 and 10 are excluded from final mean ± standard deviation evaluation as they were used in the validation set for hyperparameter tuning).**

| Metrics | Model | Rat 2 | Rat 4 | Rat 5 | Rat 6 | Rat 7 | Rat 8 | Rat 9 | Rat 10 | Mean ± Standard Deviation |
|---|---|---|---|---|---|---|---|---|---|---|
| **Accuracies** | Graph - 1D CNN | 52.74 | 59.32 | 60.04 | 75.02 | 71.70 | 61.33 | 54.32 | 60.81 | 63.62 ± 7.98 |
| | Graph – Random | 52.20 | 59.05 | 55.45 | 50.78 | 52.36 | 50.45 | 42.38 | 49.66 | 51.74 ± 5.62 |
| | Graph – Euclidean Distance | 54.21 | 56.54 | 59.49 | 68.29 | 60.57 | 67.41 | 65.09 | 51.02 | 62.90 ± 4.72 |
| **F1 Score** | Graph - 1D CNN | 45.87 | 57.10 | 51.81 | 68.67 | 67.01 | 60.11 | 49.41 | 58.51 | 59.02 ± 7.83 |
| | Graph – Random | 35.14 | 56.26 | 54.87 | 26.03 | 37.22 | 22.33 | 24.66 | 22.85 | 36.89 ± 15.35 |
| | Graph – Euclidean Distance | 41.36 | 56.87 | 56.00 | 51.69 | 50.36 | 64.32 | 58.9 | 31.67 | 56.36 ± 5.06 |

## Within-rat performance

We then evaluated the within-rat performance to establish a direct comparison with the previous study (note that the results are slightly different as one rat is removed in this study) [12]. The accuracies and F1-scores from both the model proposed in the previous study, the model proposed in this study, as well as the ablation studies performed, are

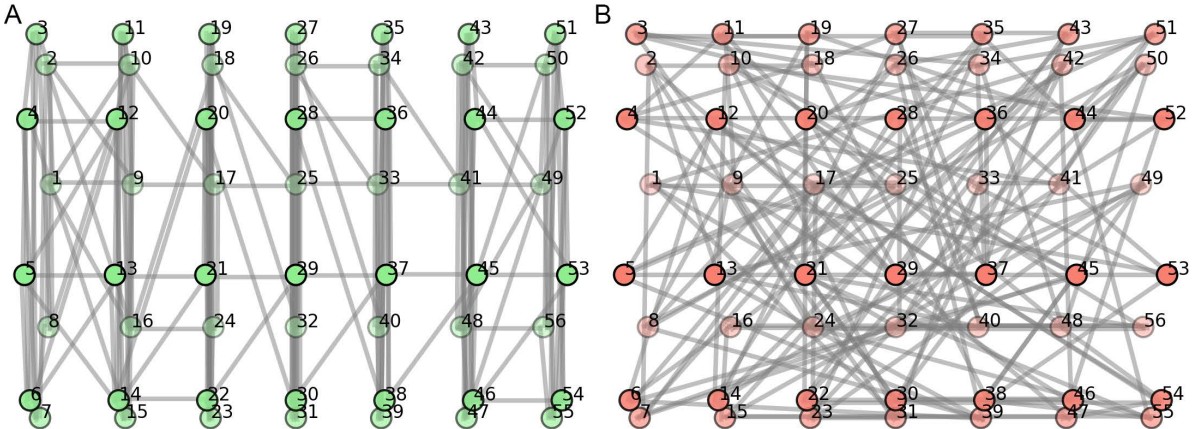

**Fig 5. Comparison of graph connectivity definitions in ablation studies.** (a) Euclidean distance–based graph. (b) Randomly constructed graph.

summarized in Table 4. The graph-based approach achieved a 1.92% improvement in accuracy and a 3.14% improvement in F1-score compared to the CNN baseline. Paired t-test analysis revealed that, although the geodesic-distance-based graph model consistently outperformed the CNN baseline, the observed improvements were not statistically significant ($p > 0.05$). In contrast, both the random graph and Euclidean-distance-based graph ablations resulted in significantly lower performance compared to the geodesic graph, indicating that preserving the physiologically meaningful geodesic structure is critical and represents the most effective choice for graph construction.

## Discussion

The proposed approach in this work showed the significance and effectiveness of using graphs to encode information from neural recordings.

The results demonstrated that the proposed graph-based learning approach outperforms conventional CNNs in classifying peripheral neural signals, both in across-subject generalization and within-subject evaluations. Compared to the reimplemented CNN baseline from [12], the graph-based model achieved substantial improvements in mean accuracy and F1 score (Tables 2 and 4), highlighting the advantage of explicitly incorporating spatial relationships between electrodes into the learning process. These gains were consistent across most test subjects, indicating robustness to subject, specific variability, a key challenge in neural decoding. It should be noted that the reported performance differences also reflect the exclusion of one rat from the analysis, which accounts for slight deviations from previously reported values [12].

Ablation studies (Tables 3 and 4) confirm that the performance advantage is not solely attributable to the LSTM module, as replacing it with a 1D CNN yielded comparable results. In contrast, removing the geodesic-distance-based graph construction and replacing it with random connectivity led to a sharp performance drop, underscoring the importance of physiologically meaningful graph structures. Using Euclidean-distance-based graphs also resulted in a noticeable reduction in performance compared to geodesic connectivity, although it consistently outperformed the random graph baseline. This pattern suggests that incorporating spatial proximity alone is beneficial, but Euclidean distances do not reflect the meaningful geometry of the nerve cuff, as they ignore the circumferential arrangement of electrodes and the way neural signals propagate along the nerve surface. In contrast, geodesic distances capture both the physical layout and the physiologically relevant pathways through which activity spreads across adjacent contacts, leading to superior performance. By modeling electrode positions using geodesic distances, the graph-based approach leverages both local and global spatial dependencies, enabling more informative spatiotemporal feature extraction than CNNs, which primarily capture local spatial patterns.

**Table 4. Within-rat performance (accuracy and macro-averaged F1 score) of baseline model, graph-based models with optimal connectivity, as well as ablation studies.**

| Metrics | Model | Rat 2 | Rat 4 | Rat 5 | Rat 6 | Rat 7 | Rat 8 | Rat 9 | Rat 10 | Mean ± Standard Deviation |
|---|---|---|---|---|---|---|---|---|---|---|
| **Accuracies** | CNN | 59.89 | 76.31 | 76.14 | 84.38 | 87.00 | 86.82 | 92.17 | 64.40 | 78.76 ± 10.41 |
| | Graph | 72.69 | 80.15 | 80.98 | 83.41 | 81.57 | 81.82 | 89.89 | 74.95 | 80.68 ± 5.66 |
| | Graph - 1D CNN | 72.54 | 78.34 | 82.21 | 84.74 | 81.82 | 81.60 | 91.76 | 72.46 | 80.68 ± 6.36 |
| | Graph – Random | 63.14 | 68.04 | 72.01 | 81.07 | 78.16 | 75.26 | 87.67 | 62.10 | 73.43 ± 8.88* |
| | Graph – Euclidean Distance | 65.29 | 70.58 | 75.33 | 81.87 | 72.28 | 73.31 | 87.29 | 66.41 | 74.04 ± 7.45* |
| **F1 Score** | CNN | 48.73 | 76.69 | 75.32 | 75.77 | 84.99 | 85.42 | 88.70 | 59.25 | 74.36 ± 13.53 |
| | Graph | 72.39 | 80.06 | 80.29 | 72.86 | 76.13 | 79.96 | 86.33 | 72.21 | 77.50 ± 5.13 |
| | Graph - 1D CNN | 71.96 | 76.50 | 81.52 | 76.26 | 76.97 | 79.97 | 87.86 | 65.34 | 77.04 ± 6.65 |
| | Graph – Random | 44.94 | 67.84 | 72.28 | 59.51 | 69.40 | 68.44 | 83.23 | 54.11 | 64.97 ± 11.81* |
| | Graph – Euclidean Distance | 49.17 | 70.24 | 73.94 | 68.65 | 54.21 | 73.03 | 77.57 | 57.36 | 65.52 ± 10.46* |

Our analysis of connectivity hyperparameters reveals that graphs constructed with more than three neighbors outperformed those with fewer, with peak performance achieved at five neighbors and a $\sigma$ of 2. This configuration appears to optimally balance local specificity and global context. The distance decay parameter plays a key role in modulating the model's sensitivity to spatial distance: higher $\sigma$ reduces the decay of edge weights with distance, allowing for more global integration when many neighbors are retained, while lower $\sigma$ enforces stronger locality. This interaction suggests that careful tuning of both parameters can maximize the model's ability to capture meaningful spatial relationships while avoiding overfitting to noise.

From a physiological perspective, the geodesic-based connectivity preserves the true spatial organization of electrodes on the nerve cuff, reflecting how neural signals propagate through the peripheral nervous system. Electrodes positioned close together are more likely to record correlated activity, while distant contacts often provide complementary, non-redundant information. Capturing these relationships is crucial for improving model performance and interpretability.

Overall, graph-based approaches not only outperformed traditional methods such as CNNs in classification performance but also have great potential in clinical explainability. One of the advantages of GNNs is their ability to provide deeper understanding into the model's decision-making process. For instance, future studies may examine the learned weights in the GNN layers to identify the most influential nodes (e.g., electrodes) and edges (e.g., connections) that contribute to the final classification decision. By investigating node importance, we can determine which specific electrode channels contributed more in distinguishing different neural patterns, providing valuable information about the distribution of neural activity. Similarly, analyzing edge importance can reveal how different electrodes interact and contribute to temporal dynamics, helping us understand how neural signals propagate across the nerve. Additionally, the graph-based approach has proven to be more generalizable, allowing for the incorporation of data from multiple subjects rather than relying solely on a single animal model for training. This ensures consistent model performance across different subjects and enhances real-world applicability.

Future studies should explore different strategies for defining graph connectivity, such as adaptive thresholding based on statistical dependencies between channels or dynamic graphs that incorporate learnable edges. Subgraph-based approaches could be investigated to focus on the most informative nodes and edges, potentially improving computational efficiency and model interpretability [36]. Additionally, self-supervised learning techniques, such as contrastive learning or graph autoencoders, could be explored to enhance the ability of the model to extract meaningful representations without relying on a large amount of labeled data. Prior studies have shown that graph-based approaches are particularly effective in low-data regimes, as they leverage relational inductive biases to learn richer representations compared to grid-based

methods [37,38]. Our findings align with this evidence, suggesting that integrating self-supervised objectives could further strengthen performance under limited data availability. A notable limitation of the present study is the relatively small dataset, consisting of recordings from only eight rats. Although statistical testing demonstrated significant performance differences between the graph-based models and the CNN baseline, the small sample size inherently limits the strength of these conclusions. This constraint reflects the practical challenges of collecting nCAPs in vivo, an experimental process that is resource-intensive and time-consuming. As larger datasets become available, future work should evaluate graph-based methods on more extensive cohorts to further validate generalizability and robustness.

While this study focuses on rat peripheral nerve recordings, the proposed graph-based framework is well positioned for translation to human applications. Human peripheral neural signals are characterized by greater anatomical variability, differences in nerve size, and increased signal heterogeneity arising from subject-specific physiology, electrode placement, and clinical noise sources. By explicitly modeling inter-electrode relationships rather than relying on fixed grid-based assumptions, the graph formulation provides a flexible representation that can naturally adapt to these sources of variability.

## Conclusion

We explored a graph-based approach for classifying three afferent activities using neural recordings obtained from Long-Evan rats with a 56-channel nerve cuff electrode. The GNN model effectively captured temporal patterns through nodal features, and by constructing weighted graph adjacency matrices with geodesic distance, the model was able to extract more informative and temporal and spatial features, outperforming the CNN model. Under the leave-one-out strategy, the GNN model outperformed the CNN model by 14.32% in mean classification accuracy, and by 17.74% in macro-averaged F1 score, and in the within-rat evaluation, the proposed model in this study achieved an improvement of 1.92% in accuracy and 3.14% in F1 score. In summary, our findings highlight the potential of graph-based models for decoding neural signals by effectively utilizing both temporal and spatial relationships. A particularly promising future direction is the interpretability of learned graph representations, which may provide insight into how individual electrodes and their interactions contribute to classification decisions. Analyzing node- and edge-level importance within the learned graphs could help reveal physiologically meaningful patterns of neural activity and improve transparency for neuroscience and clinical applications. Future work in this area can further refine graph-based approaches to enhance classification accuracy and interpretability, and expand the applicability of these models to other neural decoding tasks.

## Author contributions

**Conceptualization:** Rui Qi Ji, Mehdy Dousty, Ryan G. L. Koh.

**Data curation:** Ryan G. L. Koh.

**Formal analysis:** Rui Qi Ji, Mehdy Dousty.

**Investigation:** Rui Qi Ji, Mehdy Dousty, Ryan G. L. Koh.

**Methodology:** Rui Qi Ji, Mehdy Dousty, Ryan G. L. Koh.

**Resources:** Ervin Sejdić.

**Software:** Rui Qi Ji, Mehdy Dousty.

**Supervision:** Mehdy Dousty, Ryan G. L. Koh, Ervin Sejdić.

**Validation:** Rui Qi Ji.

**Visualization:** Rui Qi Ji.

**Writing – original draft:** Rui Qi Ji.

**Writing – review & editing:** Mehdy Dousty, Ryan G. L. Koh, Ervin Sejdić.

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
