## [Decision Letter · Decision Letter 0]

30 Nov 2025

Dear Dr. Ji,

Data leakage in hyperparameter tuning. The authors report that hyperparameter optimisation (e.g., σ, k) was performed on data later used for evaluation, risking inflated performance claims. Possibly due to a lack of critical details (see next point), it is hard to gauge to what extent this poses a risk of data leakage. On one hand a clear separation between training, model selection / parameter tuning and test datasets is a pillar of any proper model evaluation procedure. On the other hand, the authors provide a heatmap of hyperparameter performance (Figure 4) and the initial tuning step might have been used only to identify a reasonable parameter values from which to explore other combinations. It is essential that the authors clarify this point unequivocally and convincingly. Please describe the training/evaluation split and procedure step-by-step.Insufficient method details. Critical details (e.g., edge feature definitions, exact GNN formulations, code availability) are omitted, hindering reproducibility. Clarify role of "k". Equations and pseudocode should be added where needed. Consider sharing code unless not doing so is justifiable.Inappropriate data augmentation. The listed augmentations are intended for image data and not meaningful for spatio-temporal neural signals. Clarify the rationale for using them. Also confirm unequivocally that any data augmentation is performed within CV, on the training data only, and not on the pooled data prior to splitting.Weak ablation design. No baseline that uses inter‑channel correlation matrices is included (e.g. a baseline graph whose edge weights are Euclidean distances). Thus the ablation does not fully isolate the benefit of the geodesic geometry.Statistical significance. Are these differences observed statistically significant? If so, provide details of test used. The overlapping standard deviations and large variances make it difficult to claim a statistically meaningful difference.Generalisation and Clinical Relevance. Discuss how the method generalises from rat to human recordings, considering signal variability and electrode configurations (e.g., larger cuffs).

We look forward to receiving your revised manuscript.

Kind regards,

Luca Citi, PhD

Academic Editor

PLOS ONE

Journal Requirements:

4. We note you have included a table to which you do not refer in the text of your manuscript. Please ensure that you refer to Table 1 in your text; if accepted, production will need this reference to link the reader to the Table.

Reviewers' comments:

Reviewer's Responses to Questions

**Comments to the Author**

1. Is the manuscript technically sound, and do the data support the conclusions?

Reviewer #1: Yes

Reviewer #2: No

Reviewer #3: Partly

2. Has the statistical analysis been performed appropriately and rigorously?

Reviewer #1: Yes

Reviewer #2: No

Reviewer #3: No

3. Have the authors made all data underlying the findings in their manuscript fully available?

Reviewer #1: Yes

Reviewer #2: No

Reviewer #3: Yes

4. Is the manuscript presented in an intelligible fashion and written in standard English?

Reviewer #1: Yes

Reviewer #2: Yes

Reviewer #3: Yes

Reviewer #1: The paper presents a solid study that applies Graph Neural Networks to classify peripheral neural recordings, integrating the geometry of the cuff electrode into the graph construction. Below, I provide some observations to improve the quality of the manuscript, listed in order of importance.

Strong:

The state of the art should be further developed. A good comparison is made with CNN, based on the results of Koh et al., but Transformers are only mentioned without a direct comparison or concrete results. Moreover, it would be helpful to include other models often used in the literature for peripheral neural recordings analysis, such as LSTM, Inception Time, or simpler approaches like SVM, for a more complete comparison and to better contextualize the results.

In the final discussion, it would be useful to add an in-depth analysis of the generalization of the approach from rat to human, considering the differences in peripheral neural signals. The analysis could explore how the methodology can be adapted to clinical settings, where signal variability is greater. Furthermore, it would be interesting to discuss how the graph structure may need to be modified based on different electrode configurations, such as cuffs with more channels or devices for larger nerve sections, to assess the robustness of the model in real clinical scenarios.

It would be helpful to add a discussion on the interpretability of the model, to help neuroscience experts better understand how the model makes decisions. A possible future development (if not already covered) could be the analysis of the connection weights in the graphs to explore the interactions between electrodes, or how the importance of individual connections contributes to the classification results. If not addressed in the paper, it might be interesting to mention this in the conclusions as a future direction.

Minor:

It would be useful to add context on the nerve interface and include an explanatory figure of the cuff electrode to facilitate understanding, especially for less experienced readers.

The results are presented in tables, but it would be more elegant and immediate to use graphs, which could make the differences between the models and various experiments more evident.

I suggest split/shortening some long sentences to make reading easier. Some punctuation errors (for example, missing puntis “.” at the end of the figures), a general check would be usefull.

Reviewer #2: In the manuscript entitled “Enhancing Generalizability in Classification of Peripheral Neural Recordings with Graph Neural Network,” the authors propose a graph-based learning framework for classifying peripheral nerve recordings. The topic is timely and relevant to the application of graph neural networks in analysis of neural signals. Nevertheless, I have several major concerns regarding the methodological design, analysis, and presentation that limit the strength of the conclusions. My comments are organised below into major and minor points:

Major

1.Data leakage in hyperparameter tuning

Two randomly selected rats are used for hyperparameter optimisation and then reincorporated into training and evaluation. This compromises the independence of test data and likely inflates reported results. These subjects should be excluded from final evaluation, or the authors should employ nested cross-validation to avoid overlap. Furthermore, it is unclear which parameters were actually tuned, as architectural settings were fixed while only σ and k were “systematically varied.”

2. Questionable performance claims

The claim that the proposed method “outperforms the CNN” is not well supported. Table 2 shows that only a subset of configurations exceed CNN accuracy or F1-score, and significance is claimed without clear justification. Reported improvements may reflect overfitting from hyperparameter sweeps rather than genuine generalisation gains. Statistical testing on n = 8 subjects is underpowered and should not be used to claim significance.

3.Within-subject results not statistically meaningful

In Table 4, within-rat results exhibit large standard deviations and overlapping confidence intervals across methods. Differences are likely random variation and should not be interpreted as real improvements.

4. Weak ablation design

The ablation comparing geodesic graphs to random graphs is insufficient to support claims about the value of geometric priors. Stronger baselines—e.g., graphs built from inter-channel correlation matrices—are needed to determine whether improvements truly stem from spatial geometry or merely from introducing structured connectivity.

5. Inappropriate data augmentation

The listed augmentations (brightness, contrast, saturation, hue perturbation) are intended for image data and not meaningful for spatio-temporal neural signals. Their use could distort physiological relationships. The authors should justify these operations or remove them entirely.

6.Unclear architectural and training hyperparameters

Hidden-unit sizes, L2 regularisation, batch size, learning rate, and dropout rate are fixed without explanation. The basis for these choices must be stated, or the parameters should be included in the tuning procedure.

7.Insufficient methodological clarity

The paper must specify the exact formulations of the edge convolution and general graph convolution layers, including equations and citations of the specific variants used. The authors should explicitly state what constitutes edge features (e.g., edge weights or other quantities).

Minor

- Clarify dataset composition and class balance; report number of samples per class

- The description of “naturally evoked compound action potentials (nCAPs)” should be expanded for a general audience. In Equation (1), variables x and y should be explicitly defined as electrode coordinates on the cuff.

- Figures and table captions should be more detailed to allow easy understanding of what’s exactly being shown

- Specify what software was used for reproducibility

Reviewer #3: Main comments:

This manuscript is generally well written and the idea of incorporating electrode geometric distances on the nerve cuff into decoding via a GNN is sensible. However, critical methodological details are missing, and some claims are not convincingly supported because of flaws in the experimental design.

1. Insufficient method details. Figure 2 describes the proposed method at a high level, but crucial details are omitted. Although EdgeConv and GeneralConv exist in literature, the authors should provide mathematical formulations (at least in the supplement) and ideally share code so readers can reproduce the work. Without these specifics, it is difficult to determine the source of the reported improvements for the GNN relative to other methods.

2. Unclear role of k. The authors state, “To refine the graph topology, we use the σ value and number of nearest neighbours k as tunable hyperparameters.” Since the adjacency matrix A is determined by σ, the role of k is ambiguous. Given that k strongly affects results, please explain precisely how k is used to construct A (pseudocode or equations in the supplement would help).

3. Inappropriate baseline in ablation. In the ablation studies (Sec. 2.5) the authors use a random graph as a baseline. To support the claim that GNNs improve over CNNs, a more appropriate baseline is a graph whose edge weights are the Euclidean distances between electrodes in the 2D plane (instead in the cylinder). This better approximates the local receptive-field scanning performed by CNNs. Please add this comparison.

4. Unexpected random-graph result requires explanation. The authors report that “Interestingly, when using a random graph in place of the correct adjacency matrix, the accuracy remained close to that of the CNN baseline. This suggests…”. This requires a deeper explanation, since it implies that CNN weights may be unimportant—counterintuitive to expectations (random < CNN < GNN). Please analyze and discuss potential reasons.

Minor comments:

1. Abstract claim on small-data regimes. The abstract claims benefits “particularly in small-data regimes where models cannot reliably learn such relationships solely from the data.” I do not see experiments testing varying dataset sizes. If this claim is retained, provide supporting experiments or tone down the claim.

2. Figure 3 / neighborhood k. In Figure 3 the reported neighborhood size k = 5 is not visually consistent (e.g., electrode 20 appears connected to only four neighbors). Please correct the figure or clarify the visualization.

3. Within-subject cross-validation details. In Sec. 2.4.2 you state “two folds for training” for within-subject evaluations. Please describe the training/evaluation split and procedure step-by-step in the supplement (fold definitions, random seeds, how validation/test sets were selected, etc.).

4. Inconsistent CNN accuracy values. In Sec. 3.1 CNN accuracy is reported as 52.00 ± 5.17%, while it’s 52.00 ± 5.86% in Table 3. Please reconcile and explain any differences.

5. Misleading axis label “temperature.” In Figure 4 the x-axis labeled “temperature” appears to represent σ. Rename the axis to σ or give it a meaningful term consistent with the graph-construction procedure. Replace “temperature” throughout the manuscript.

6. Statistical significance. In Sec. 3.2 the authors report “a 1.92% improvement in accuracy and a 3.14% improvement in F1-score compared to the CNN baseline.” Are these differences statistically significant? Are the differences in Table 4 significant? Please report appropriate statistical tests (e.g., paired tests across folds/datasets) and include p-values or confidence intervals.

.

Reviewer #1: No

Reviewer #2: **Yes:** Dominik KleplDominik KleplDominik KleplDominik Klepl

Reviewer #3: No

---

## [Author Response · Author response to Decision Letter 1]

15 Jan 2026

We sincerely thank the editor for the opportunity to revise our manuscript and for recognizing the importance of our work, and greatly appreciate the constructive feedback provided. We have considered each comment and have made revisions to the manuscript to address all concerns raised. Responses to each reviewer comment are provided in blue text throughout this letter.

Editor’s comments:

1. Data leakage in hyperparameter tuning. The authors report that hyperparameter optimisation (e.g., σ, k) was performed on data later used for evaluation, risking inflated performance claims. Possibly due to a lack of critical details (see next point), it is hard to gauge to what extent this poses a risk of data leakage. On one hand a clear separation between training, model selection / parameter tuning and test datasets is a pillar of any proper model evaluation procedure. On the other hand, the authors provide a heatmap of hyperparameter performance (Figure 4) and the initial tuning step might have been used only to identify a reasonable parameter values from which to explore other combinations. It is essential that the authors clarify this point unequivocally and convincingly. Please describe the training/evaluation split and procedure step-by-step.

We thank the editor for raising this point. In the revised manuscript, we have fully addressed the concern regarding hyperparameter tuning and data leakage. Specifically, the two rats used for hyperparameter optimization (Rats 2 and 10) are now excluded from all final performance evaluations in Tables 2 and 3, ensuring a strict separation between validation and test data. We also clarify exactly which hyperparameters were tuned, training parameters (learning rate, batch size, L2 penalty), architectural parameters (hidden units, dropout), as well as graph construction parameters (number of neighbours and distance-scaling parameter) in Methods, subsection Training & Evaluation, Hyperparameter Tuning (pages 6-7). To clarify further, the heatmap in Figure 4 is provided specifically to illustrate how different graph topologies influence performance. The purpose of this analysis is to demonstrate the importance of graph topology in problems where physiologically informed data structure matters, and the variation across σ and k confirms the critical role of incorporating meaningful nerve geometry.

For a more detailed training/evaluation procedure, we have shared our graph construction and model training code on GitHub: https://github.com/ruiqi1124/graph_nerve_recording.git

2. Insufficient method details. Critical details (e.g., edge feature definitions, exact GNN formulations, code availability) are omitted, hindering reproducibility. Clarify role of "k". Equations and pseudocode should be added where needed. Consider sharing code unless not doing so is justifiable.

We thank the editor for raising this point, we have now expanded our Methods, subsection Model Architecture (pages 5 – 6) to elaborate on the mathematical formulations of the GNN layers.

3. Inappropriate data augmentation. The listed augmentations are intended for image data and not meaningful for spatio-temporal neural signals. Clarify the rationale for using them. Also confirm unequivocally that any data augmentation is performed within CV, on the training data only, and not on the pooled data prior to splitting.

We have now clarified this in the manuscript, in Methods, subsection Model Architecture (last paragraph, page 6). We also confirm that any data augmentation was only performed during model training, on the training set.

4. Weak ablation design. No baseline that uses inter channel correlation matrices is included (e.g. a baseline graph whose edge weights are Euclidean distances). Thus the ablation does not fully isolate the benefit of the geodesic geometry.

We have included a third ablation study, using graphs constructed with Euclidean distances. The description has been added to Methods, subsection Ablation Studies (pages 7-8), and the results are presented in Tables 3 and 4. More relevant discussion has also been added in the Discussion section (page 11).

5. Statistical significance. Are these differences observed statistically significant? If so, provide details of test used. The overlapping standard deviations and large variances make it difficult to claim a statistically meaningful difference.

Paired t-tests were used in comparing results in Tables 2 and 3, and any statistical significance is commented on in the Results section (subsection Generalizability Performance, page 8 for Table 2, and subsection Within-Rat Performance, page 10 for Table 3).

6. Generalisation and Clinical Relevance. Discuss how the method generalises from rat to human recordings, considering signal variability and electrode configurations (e.g., larger cuffs).

We have added a paragraph on this in the Discussion section (last paragraph, page 12).

Reviewers’ Comments

We sincerely thank the reviewers for their time, thoughtful feedback, and careful evaluation of the initial version of our manuscript. We have carefully considered the comments and have revised the manuscript accordingly and added further details and clarifications to address the concerns raised.

Reviewer #1: The paper presents a solid study that applies Graph Neural Networks to classify peripheral neural recordings, integrating the geometry of the cuff electrode into the graph construction. Below, I provide some observations to improve the quality of the manuscript, listed in order of importance.

We thank the reviewer for taking the time to provide us with constructive feedback. Additional explanations were added to relevant sections as emphasized below.

Strong:

1. The state of the art should be further developed. A good comparison is made with CNN, based on the results of Koh et al., but Transformers are only mentioned without a direct comparison or concrete results. Moreover, it would be helpful to include other models often used in the literature for peripheral neural recordings analysis, such as LSTM, Inception Time, or simpler approaches like SVM, for a more complete comparison and to better contextualize the results.

We revised the Introduction section to provide a broader overview of existing methods (third paragraph, page 2). We expanded the comparison to explicitly discuss other deep learning techniques and machine learning classifiers for peripheral nerve decoding. We still believe that CNN is the most meaningful comparison with graph-based models, as no other models can encode the spatial relationship, and the geometric topology of the nerve cuff. We have highlighted this point even more in the introduction, and added a new citation 20 to support our point. We also added rationale for why these approaches cannot encode cuff geometry.

Additionally, we have clarified in the Introduction section that CNN and LSTM are indeed compared, in ablation studies.

2. In the final discussion, it would be useful to add an in-depth analysis of the generalization of the approach from rat to human, considering the differences in peripheral neural signals. The analysis could explore how the methodology can be adapted to clinical settings, where signal variability is greater. Furthermore, it would be interesting to discuss how the graph structure may need to be modified based on different electrode configurations, such as cuffs with more channels or devices for larger nerve sections, to assess the robustness of the model in real clinical scenarios.

We added a new paragraph in the Discussion section explicitly analyzing translation from rat to human recordings (last paragraph, page 12). The revised text discusses differences in fascicle size, nerve diameter, and signal variability, and addresses how graph structure may change for cuffs with different channel densities or geometries. We also outline how the approach can adapt to clinical nerve-interface designs.

3. It would be helpful to add a discussion on the interpretability of the model, to help neuroscience experts better understand how the model makes decisions. A possible future development (if not already covered) could be the analysis of the connection weights in the graphs to explore the interactions between electrodes, or how the importance of individual connections contributes to the classification results. If not addressed in the paper, it might be interesting to mention this in the conclusions as a future direction.

We thank the reviewer for mentioning this important direction for future work. We have elaborated on this in the third last paragraph in the Discussion section (page 11).

Minor:

It would be useful to add context on the nerve interface and include an explanatory figure of the cuff electrode to facilitate understanding, especially for less experienced readers.

The results are presented in tables, but it would be more elegant and immediate to use graphs, which could make the differences between the models and various experiments more evident.

In response to the points above, we have added figures illustration graph connectivity using geodesic distance, Euclidean distance, and random graph construction (Figures 3, page 7, and 5, page 10). These provide more context for the models described in the tables. The geodesic distance-based graph is the main proposed model of this study, while the other ones are part of ablation studies.

I suggest split/shortening some long sentences to make reading easier. Some punctuation errors (for example, missing puntis “.” at the end of the figures), a general check would be useful.

We thank the reviewers for pointing this out. We have revised this accordingly.

Reviewer #2: In the manuscript entitled “Enhancing Generalizability in Classification of Peripheral Neural Recordings with Graph Neural Network,” the authors propose a graph-based learning framework for classifying peripheral nerve recordings. The topic is timely and relevant to the application of graph neural networks in analysis of neural signals. Nevertheless, I have several major concerns regarding the methodological design, analysis, and presentation that limit the strength of the conclusions. My comments are organised below into major and minor points:

We thank the reviewer for taking the time to provide us with constructive feedback. Additional explanations were added to relevant sections as emphasized below.

Major

1.Data leakage in hyperparameter tuning

Two randomly selected rats are used for hyperparameter optimisation and then reincorporated into training and evaluation. This compromises the independence of test data and likely inflates reported results. These subjects should be excluded from final evaluation, or the authors should employ nested cross-validation to avoid overlap. Furthermore, it is unclear which parameters were actually tuned, as architectural settings were fixed while only σ and k were “systematically varied.”

We thank the reviewer for requesting this clarification and agree that it is essential for transparency and reproducibility. In the revised manuscript, Rats 2 and 10, which were used for hyperparameter tuning, have been excluded from all final performance metrics (mean ± standard deviation) in Tables 2 and 3 to prevent data leakage in the generalizability analysis. Their individual results remain in the tables solely for contextual reference. Statistical analyses (paired t-tests) are now conducted only on the remaining rats, and we have clearly specified the hyperparameters tuned in Methods, subsection Training & Evaluation, Hyperparameter Tuning (pages 6 – 7).

2. Questionable performance claims

The claim that the proposed method “outperforms the CNN” is not well supported. Table 2 shows that only a subset of configurations exceed CNN accuracy or F1-score, and significance is claimed without clear justification. Reported improvements may reflect overfitting from hyperparameter sweeps rather than genuine generalisation gains. Statistical testing on n = 8 subjects is underpowered and should not be used to claim significance.

We acknowledge that the sample size is small and agree that this is an inherent limitation of the study. Unfortunately, in vivo peripheral nerve recording experiments are technically demanding and resource-intensive, and datasets of this modality are scarce in the literature. We now explicitly describe this limitation in the second-to-last paragraph of the Discussion (page 12). Importantly, after addressing the first major revision point and excluding the two rats used for hyperparameter tuning, the graph-based approach with 4 or 5 neighbors still significantly outperforms the CNN baseline in generalizability (p < 0.05, paired t-test). These results have been updated and clarified in Table 2 and in subsection Generalizability Performance of the Results (page 8).

3.Within-subject results not statistically meaningful

In Table 4, within-rat results exhibit large standard deviations and overlapping confidence intervals across methods. Differences are likely random variation and should not be interpreted as real improvements.

We have now clarified this in Results, subsection Within-Rat Performance (page 10), expanding more on the performance results in Table 4.

4. Weak ablation design

The ablation comparing geodesic graphs to random graphs is insufficient to support claims about the value of geometric priors. Stronger baselines—e.g., graphs built from inter-channel correlation matrices—are needed to determine whether improvements truly stem from spatial geometry or merely from introducing structured connectivity.

We added a Euclidean-distance graph baseline, generated using inter-electrode 2D planar distances. Results are presented in updated Tables 3 and 4. We discuss how Euclidean graphs perform better than random graphs but worse than geodesic graphs, supporting the claim that true anatomical geometry provides additional benefit (third paragraph of Discussion section, page 11).

5. Inappropriate data augmentation

The listed augmentations (brightness, contrast, saturation, hue perturbation) are intended for image data and not meaningful for spatio-temporal neural signals. Their use could distort physiological relationships. The authors should justify these operations or remove them entirely.

We have now clarified this and removed any inappropriate augmentations described. The final augmentation pipeline uses only low-amplitude Gaussian noise, which is physiologically meaningful for neural signals. This is described clearly in the revised Methods, subsection Model Architecture (last paragraph, page 6).

6.Unclear architectural and training hyperparameters

Hidden-unit sizes, L2 regularisation, batch size, learning rate, and dropout rate are fixed without explanation. The basis for these choices must be stated, or the parameters should be included in the tuning procedure.

We have now clarified this in subsection Training & Evaluation, Hyperparameter Tuning of Methods (page 7).

7.Insufficient methodological clarity

The paper must specify the exact formulations of the edge convolution and general graph convolution layers, including equations and citations of the specific variants used. The authors should explicitly state what constitutes edge features (e.g., edge weights or other quantities).

We thank the reviewer for asking for clarification on formulations of the convolutional layers. These have now been incorporated into Methods, subsection Model Architecture (pages 5 – 6).

Minor

- Clarify dataset composition and class balance; report number of samples per class

- The description of “naturally evoked compound action potentials (nCAPs)” should be expanded for a general audience. In Equation (1), variables x and y should be explicitly defined as electrode coordinates on the cuff.

- Figures and table captions should be more detailed to allow easy understanding of what’s exactly being shown

- Specify what software was used for reproducibility

We thank the reviewer for pointing these out, and we have addressed each item in the revised manuscript. Dataset composition has been updated and is now summarized in Table 1. The description of naturally evoked CAPs has been clarified in Methods, subsection Data Des

---

## [Decision Letter · Decision Letter 1]

24 Feb 2026

Dear Dr. Ji,

Thank you for submitting your manuscript to PLOS ONE. After careful consideration, we feel that the paper is close to meeting PLOS ONE’s publication criteria but we would like you to consider some final suggestions. Therefore, we invite you to submit a revised version of the manuscript that considers the points raised during the review process.

We look forward to receiving your revised manuscript.

Kind regards,

Luca Citi, PhD

Academic Editor

PLOS One

**Journal Requirements:**

Reviewers' comments:

Reviewer's Responses to Questions

**Comments to the Author**

Reviewer #1: All comments have been addressed

Reviewer #3: All comments have been addressed

2. Is the manuscript technically sound, and do the data support the conclusions?

Reviewer #1: Yes

Reviewer #3: Yes

3. Has the statistical analysis been performed appropriately and rigorously?

Reviewer #1: Yes

Reviewer #3: Yes

4. Have the authors made all data underlying the findings in their manuscript fully available?

Reviewer #1: Yes

Reviewer #3: Yes

5. Is the manuscript presented in an intelligible fashion and written in standard English?

Reviewer #1: Yes

Reviewer #3: Yes

Reviewer #1: I would like to thank the authors for their comprehensive responses and the revisions made to the manuscript. The clarifications provided are exhaustive and satisfactorily address my previous concerns regarding the state-of-the-art comparison and the methodological details.

I have just one minor note regarding the presentation of the results, which is a suggestion and not a mandatory requirement for acceptance. While I appreciate the inclusion of the network topology figures, I believe that the quantitative results currently presented in Tables 2, 3, and 4 would be much more effective if visualized as charts (e.g., grouped bar plots with error bars). Visualizing the data in this way would allow readers to immediately grasp the performance trends and the stability of the model across subjects, whereas large tables can be harder to interpret at a glance. The detailed numerical tables could then be moved to the Supplementary Material.

However, I leave this decision entirely to the authors' discretion. I am satisfied with the scientific content and the improvements made, and I support the publication of the manuscript in its current form.

Reviewer #3: All my comments have been addressed properly. I don't have further questions. The comparison between geodesic-based and the Euclidean graph looks reasonable, which I think is the key of this paper.

.

Reviewer #1: No

Reviewer #3: No

---

## [Author Response · Author response to Decision Letter 2]

2 Mar 2026

We sincerely thank the editor and reviewers for their time in reviewing the revised manuscript, and greatly appreciate the constructive feedback provided. We have considered each comment. Responses to each reviewer comment are provided in blue text throughout this letter.

Reviewer #1: I would like to thank the authors for their comprehensive responses and the revisions made to the manuscript. The clarifications provided are exhaustive and satisfactorily address my previous concerns regarding the state-of-the-art comparison and the methodological details.

I have just one minor note regarding the presentation of the results, which is a suggestion and not a mandatory requirement for acceptance. While I appreciate the inclusion of the network topology figures, I believe that the quantitative results currently presented in Tables 2, 3, and 4 would be much more effective if visualized as charts (e.g., grouped bar plots with error bars). Visualizing the data in this way would allow readers to immediately grasp the performance trends and the stability of the model across subjects, whereas large tables can be harder to interpret at a glance. The detailed numerical tables could then be moved to the Supplementary Material.

However, I leave this decision entirely to the authors' discretion. I am satisfied with the scientific content and the improvements made, and I support the publication of the manuscript in its current form.

We sincerely thank the reviewer for their kind remarks and thoughtful feedback on the revised manuscript. We appreciate the suggestion to visualize the quantitative results in Tables 2, 3, and 4 using grouped bar plots with error bars. After careful consideration, we chose to retain the tables in the main text because a few of the reported differences between models may be less clearly distinguishable when visualized in plot form, and we believe that presenting the exact numerical values allows for more precise comparison. We are grateful for the reviewer’s support and constructive suggestions that have helped improve the quality and clarity of the manuscript.

Reviewer #3: All my comments have been addressed properly. I don't have further questions. The comparison between geodesic-based and the Euclidean graph looks reasonable, which I think is the key of this paper.

We sincerely thank the reviewer for their time and thoughtful feedback on the revised manuscript. We greatly appreciate the careful evaluation and are pleased that the comparison between geodesic-based and Euclidean graph constructions is now clear and reasonable. The reviewer’s comments have been instrumental in strengthening the manuscript, and we are grateful for their support.

---

## [Editor Report · Decision Letter 2]

3 Mar 2026

Enhancing Generalizability in Classification of Peripheral Neural Recordings with Graph Neural Network

PONE-D-25-46080R2

Dear Dr. Ji,

We’re pleased to inform you that your manuscript has been judged scientifically suitable for publication and will be formally accepted for publication once it meets all outstanding technical requirements.

Kind regards,

Luca Citi, PhD

Academic Editor

PLOS One
---

## [Editor Report · Acceptance letter]

PONE-D-25-46080R2

PLOS One

Dear Dr. Ji,

I'm pleased to inform you that your manuscript has been deemed suitable for publication in PLOS One. Congratulations! Your manuscript is now being handed over to our production team.

Kind regards,

on behalf of

Dr. Luca Citi

Academic Editor

PLOS One